# Development of a Selective Tumor-Targeted Drug Delivery System: Hydroxypropyl-Acrylamide Polymer-Conjugated Pirarubicin (P-THP) for Pediatric Solid Tumors

**DOI:** 10.3390/cancers13153698

**Published:** 2021-07-23

**Authors:** Atsushi Makimoto, Jun Fang, Hiroshi Maeda

**Affiliations:** 1Department of Hematology/Oncology, Tokyo Metropolitan Children’s Medical Center, Tokyo 183-8561, Japan; 2Faculty of Pharmaceutical Science, Sojo University, Kumamoto 860-0082, Japan; fangjun@ph.sojo-u.ac.jp; 3BioDynamics Research Foundation, Kumamoto 862-0954, Japan; maedabdr@sweet.ocn.ne.jp; 4Department of Microbiology, Kumamoto University School of Medicine, Kumamoto 862-0954, Japan; 5Tohoku University, Miyagi 980-8572, Japan; 6Faculty of Medicine, Osaka University, Osaka 565-0871, Japan

**Keywords:** enhanced permeability and retention effect, EPR effect, hydroxypropyl acrylamide polymer-conjugated pirarubicin, P-THP, anthracyclines, nanomedicine, drug delivery system, DDS, targeted drug delivery, pediatric cancers

## Abstract

**Simple Summary:**

Hydroxypropyl acrylamide polymer-conjugated pirarubicin (P-THP), an innovative polymer-conjugated anticancer agent, theoretically has highly tumor-specific distribution via the enhanced permeability and retention (EPR) effect. While anthracyclines are extremely important in the treatment of most pediatric solid tumors, P-THP may serve as a less toxic and more effective substitute for conventional anthracyclines in both newly diagnosed and refractory/recurrent pediatric cancers.

**Abstract:**

Most pediatric cancers are highly chemo-sensitive, and cytotoxic chemotherapy has always been the mainstay of treatment. Anthracyclines are highly effective against most types of childhood cancer, such as neuroblastoma, hepatoblastoma, nephroblastoma, rhabdomyosarcoma, Ewing sarcoma, and so forth. However, acute and chronic cardiotoxicity, one of the major disadvantages of anthracycline use, limits their utility and effectiveness. Hydroxypropyl acrylamide polymer-conjugated pirarubicin (P-THP), which targets tumor tissue highly selectively via the enhanced permeability and retention (EPR) effect, and secondarily releases active pirarubicin molecules quickly into the acidic environment surrounding the tumor. Although, the latter rarely occurs in the non-acidic environment surrounding normal tissue. This mechanism has the potential to minimize acute and chronic toxicities, including cardiotoxicity, as well as maximize the efficacy of chemotherapy through synergy with tumor-targeting accumulation of the active molecules and possible dose-escalation. Simply replacing doxorubicin with P-THP in a given regimen can improve outcomes in anthracycline-sensitive pediatric cancers with little risk of adverse effects, such as cardiotoxicity. As cancer is a dynamic disease showing intra-tumoral heterogeneity during its course, continued parallel development of cytotoxic agents and molecular targeting agents is necessary to find potentially more effective treatments.

## 1. Introduction

The maximum effectiveness with minimum toxicity describes the ideal anti-cancer drug. Recent developments in cancer genetics and molecular biology have accelerated the development of therapeutics away from cytotoxic agents to molecular targeting agents. The number of late-stage pipeline therapies has grown from 481 in 2008 to 849 in 2018 for a total increase of 77%. In contrast, only 62 (7.3%) cytotoxic agents were being developed in 2018 [1]. However, conventional cytotoxic agents, such as alkylating agents and anthracyclines, are still the mainstay of multidisciplinary treatment for pediatric cancers even in the era of precision medicine [2]. In this regard, improving the drug-delivery system (DDS) for cytotoxic agents can help reducing their toxicity and enable increased dose intensity. Therefore, further research and development of more effective DDS are warranted.

Hydroxypropyl methacrylate (HPMA) polymer-conjugated pirarubicin (P-THP), an innovative polymer-conjugated anticancer drug, has highly tumor-specific distribution owing to the enhanced permeability and retention (EPR) effect [3,4]. The tumor-targeting EPR effect of macromolecules was originally described in solid tumors by Matsumura and Maeda (a coauthor of the present article) in 1986 [5]. The aberrant architecture of tumorous blood vessels, active production of various vascular permeability factors, and lack of lymphatic drainage in tumor tissue, constitute the tumor-specific conditions necessary for the EPR effect [3,5]. Although, the mechanism of P-THP is often confused with that of other nanomedicine agents, liposomal doxorubicin (Doxil^®^; Janssen Pharmaceuticals, Beerse, Belgium) [6], which has been approved for clinical use in several nations, and possesses distinct pharmacodynamic characteristics that set it apart from liposomal doxorubicin. In the present review, the authors explain the mechanism underlying the high tumor-selectivity of P-THP and the application of this agent to a wide variety of pediatric malignancies.

## 2. EPR Effect and Its Mechanism

### 2.1. Discovery of the EPR Effect

The EPR effect was first observed in the retention of Evans blue-albumin complex (EAC) in S-180 tumors in mice after intravenous injection via the tail vein. Quantification of EAC in the tumors and other organs revealed that the EAC concentration was about 10-fold higher in tumor tissue than in blood at 145 h after injection [5]. Subsequently, accumulation of the macromolecules in the tumor site was confirmed through an experiment using radiolabeled serum proteins with various molecular weights (MW), including IgG (170 kDa), transferrin (90 kDa), albumin (67 kDa), and ovalbumin (48 kDa). On the other hand, low MW proteins, such as ovomucoid (29 kDa) and neocarzinostatin (12 kDa), did not accumulate in the tumor or other organs [5]. This phenomenon was dubbed the “EPR effect”.

To confirm the MW dependency of the EPR effect, biocompatible synthetic copolymers of HPMA, which can be synthesized to have various MW ranging from 4.5 to 800 kDa, were used. Repetition of the experiment described above using S-180 bearing mice with intravenous administration of radioiodinated HPMA found that the EPR effect occurred only when molecules with MW > 40 kDa were used. Although all the HPMA copolymers accumulated in the tumor regardless of MW (1.0–1.5% of the injected dose per gram of tumor) within ten minutes after injection, only copolymers with MW > 40 kDa showed significantly higher intratumor accumulation after six hours [7]. Blood clearance was slower with high MW copolymers, and the tissue levels were consistently 3–5% dose/gm kidney in the early phase, but their accumulation in the kidneys and liver was not time-dependent [7]. The EPR effect in solid tumors appeared to arise primarily from the difference in clearance rates between the solid tumors and the normal tissues after the initial penetration of the polymers into these tissues.

Evidence of the EPR effect in clinical practice can be observed via angiography of liver tumors using a lipid contrast agent (Lipiodol^®^; Guerbet LLC, Princeton, NJ, USA) administered intraarterially [8] and via gallium scintigraphy using radioactive gallium-transferrin complex (90 kDa), which accumulates in tumors and is therefore useful for their diagnosis [9]. As discussed later in this article, the EPR effect is advantageous not only for diagnosis, but also for therapy because of this preferential retention of macromolecular drugs.

### 2.2. Anatomical, Physiological, and Biochemical Basis of the EPR Effect

The EPR effect reflects several, unique, vascular properties in tumor tissue having anatomical, physiological, and biochemical aspects.

The anatomical aspect is illustrated in Figure 1. Blood vessels in the tumor (Figure 1E–H) are structurally abnormal, lacking pericytes (smooth muscle layer) and dilating and constricting with irregular diameters and aberrant branching (Figure 1G) [10,11]. They possess large pores and endothelial gap junctions as large as 600–800 nm (Figure 1E,H), which lead to extravasation of intravenously injected, high MW, polymeric resin (Figure 1E,F) [12,13]. This microarchitecture of the blood vessels can be visualized clearly by scanning electron microscopy (SEM), which reveals a significant contrast with the vasculature of normal tissue (Figure 1A–D) [10,11,12,13,14].

The excessive production of mediators, including bradykinin [14,15,16,17,18], nitric oxide (NO) [19], prostaglandins (PGs) [20], and vascular endothelial growth factor (VEGF) [21], also contributes to the hyperpermeability of tumor blood vessels as well as host vessels at the periphery of tumors (Figure 2). Maeda et al. demonstrated that the kallikrein-kinin cascade, which commonly occurs in the inflammation process, can be triggered by exogenous proteases regardless of whether they originated in bacteria or tumor cells [15,22]. Kinin causes not only the clinical symptom of pain but also leads to peritoneal and pleural effusion secondary to the enhanced permeability of the blood vessels [14,15,16,17,18]. Angiotensin converting enzyme (ACE) inhibitors, such as enalapril, increases drug delivery approximately 2-to 3-fold probably by inhibiting kinin degradation, which results in enhanced vascular permeability.

Viral and bacterial infections induce reactive oxygen species, such as superoxide anion radicals (O_2_·^−^) and reactive nitrogen species, such as NO and peroxynitrite (ONOO^−^). Solid tumor cells highly express NO synthase (NOS) and both NO and ONOO^−^ possibly enhance vascular permeability in solid tumors by activating matrix metalloproteinase [22,23]. The process is enhanced by PGs, VEGF, and inflammatory cytokines, which activate NOS to increase NO production [15,20,21,24]. As NO plays an important role in vascular permeability, use of nitroglycerin, which increases NO production, possibly enhances the EPR effect and improves macromolecular drug delivery to the tumor [25].

Last, tumor-associated lymphatic vessels also show irregularities in structure, with some tumors showing a complete lack of lymphatics. Drainage has therefore been found to be impaired in tumors [5,7,26], contributing to prolonged drug retention within the tumors.

## 3. Design of Tumor-Specific Drug Delivery Utilizing the EPR Effects

A recent report by the multinational European Technology Platform on Nanomedicine stated that “the nanomedicine field is concretely able to design products that overcome critical barriers in conventional medicine in a unique manner” [27]. However, there is some skepticism about the effectiveness of nanomedicines owing to past cases of failure at the development stage [28]. In order to maximize the effectiveness of nanomedicines, the following three conditions at least must be met: (1) Selective accumulation of the macromolecule in the tumor; (2) release of active pharmaceutical ingredients (API) into the tumor tissue; and (3) active cellular uptake of the API into the tumor cells [29]. Figure 3 illustrates how these conditions are necessary for the nanomedicine to target tumors selectively.

### 3.1. Selective Accumulation of the Nanoparticle in the Tumor

There are several requirements in designing a nanoparticle meeting the first criterion. As a minimal requirement, there should be no interaction with blood components or blood vessels, no antigenicity, no clearance by the reticuloendothelial system, and no cell lysis. Only when these conditions are satisfied can the next three factors be considered.

First, a sufficient concentration of the nanoparticle needs to be maintained in the blood stream for several hours to exert the EPR effect, resulting in selective accumulation of the drug in the tumor [7,30]. The stability of the nanoparticle is necessary for maintaining a sufficient half-life. Most non-covalently connected micelles (NCCMs) are very unstable in the blood stream; block copolymer micelle carriers containing doxorubicin, such as NK911 (Nippon Kayaku Co., Ltd.; Tokyo, Japan), for instance, have a very short plasma half-life (t_1/2_) of less than three hours in humans, which is thought to be the reasons for its ineffectiveness [31,32].

Second, the nanoparticles must be larger than 40 kDa to prevent their excretion via the kidneys. HPMA polymer-conjugated doxorubicin, which is very similar in design to P-THP, failed to produce a good antitumor effect in past studies [33] probably because, among other possible reasons, the HPMA polymer was too small (20–30 kDa) to produce the EPR effect.

Third, the electric charge of the particle surface should be neutral or weakly negative. As the vascular endothelial surface dense is generally negatively charged, particles with a positive charge will easily stick to reticuloendothelial cells in the blood vessels, resulting in a short t_1/2_ [34]. Moreover, particles with a negative charge also tend to become trapped by reticuloendothelial cells, resulting in their accumulation in the spleen and liver [34,35].

### 3.2. Release of the API in the Tumor Tissue

Although stability is one of the key factors in producing the EPR effect as described above, excessive stability is not desirable for releasing the APIs into the tumor site. If micellar or liposomal drugs are too stable, they may not release the APIs from complexes or nanomedicines even if the EPR effect delivers them to the tumor site. Doxil^®^, which has a surface coating of polyethylene glycol (PEG), releases its API so slowly that it failed to achieve clinical efficacy even in doxorubicin-sensitive tumors [6]. In contrast, NK911, as described above, is rather unstable in the blood stream and released 50% of the APIs within two hours and 100% within 24 h [31,32]. Consequently, NK911 can achieve an intra-tumoral concentration of doxorubicin only twice as high as that of doxorubicin alone [31,32].

To design a nanoparticle which is capable of releasing the APIs into the tumor tissue, tumor-specific conditions may be exploited to cleave the bond connecting the APIs to the particle. One possibility is using a peptide-linker cleavable by cathepsin B, which is highly expressed in various tumor cells [36]. Another possibility is using acid-cleavable linkages, such as the hydrazone-bond, which was used with P-THP, as will be shown in the following sections.

### 3.3. Active Cellular Uptake of APIs in Tumor Cells

Most cytotoxic agents must pass through the tumor cell membrane and interact with the DNA or organelles, in order to exert their cytotoxic function. As cellular endocytosis of a macromolecule, including nanoparticles, is usually not as efficient as that of a low-MW agent, identifying a method for the release of an API that can actively bind with, and be internalized in tumor cells, is a crucial concern in nanomedicine. The effectiveness of cell binding and internalization can be improved by adding or substituting certain residues. An example is 4′-*O*-tetrahydropyranyldoxorubicin or pirarubicin, a derivative of doxorubicin commercially available in the EU, Japan, and other Asian nations [37]. Pirarubicin is less cardiotoxic than doxorubicin [38], has an efficacy profile comparable to that of doxorubicin, and has demonstrated efficacy against ovarian cancer [39], breast cancer [40], and pediatric tumors, such as neuroblastoma [41,42], hepatoblastoma [43], and rhabdomyosarcoma [44].

Pirarubicin is taken up more rapidly into tumor cells in higher intracellular concentrations than doxorubicin. The superior cellular uptake of pirarubicin may be attributed to the pyranose residue, whose structure is similar to that of glucose. Pyranose is taken up via the glucose transporter system, which is highly upregulated in tumor cells. Pirarubicin, but not doxorubicin, is also taken up via concentrative nucleoside transporter 2 (CNT-2), which is highly expressed in tumor cells [45]. The superiority of the cellular uptake of pirarubicin to that of doxorubicin possibly results in stronger antitumor activity both in vitro and in vivo [46,47,48]. The rapid intracellular uptake of free pirarubicin continues even after its conjugation with the HPMA polymer. Whereas, the doxorubicin-polymer conjugate shows extremely poor cellular uptake and poor biological activity [49].

## 4. Designing an Ideal Nanomedicine: Our Experience Developing P-THP

P-THP is an innovative, polymer-conjugated anticancer drug, which satisfies the three criteria described in the previous section [29]. Although many candidate nanodrugs are being developed, very few nanomedicines for cancer chemotherapy fulfill all the requirements for clinical use, including batch-to-batch reproducibility, long-term stability, complexity of the manufacturing processes, and maintenance of sterile conditions [50]. Thus far, P-THP appears to meet these requirements and is awaiting regulatory clearance for testing in a clinical trial. In the following sections, we will explain the process of designing nanomedicines by describing our experience of developing P-THP.

### 4.1. Selecting Materials for the Carrier, APIs, and the Link between Them

The carrier molecule in nanomedicine must be nontoxic, non-immunogenic, and highly biocompatible. The HPMA polymer was chosen because it can be manufactured easily, is highly reproducible, and has a readily controlled MW. High MW HPMA polymers (more than 40 kDa) have been shown to be stable in the systemic circulation over long periods and to accumulate preferentially in tumor tissue via the EPR effect [7].

Pirarubicin was chosen as the API because it has various favorable characteristics, including an excellent antitumor effect with broad-spectrum sensitivity, rapid cellular uptake, and the ability to achieve a high intracellular concentration, as described earlier. Then, the hydrazone bond, an acid-cleavable linkage, was used to connect these two molecules to ensure release of the API into the tumor site.

Figure 4 shows the chemical structure of P-THP. The MW of the carrier HPMA polymer was 38,500. The pirarubicin loading in P-THP is 8.6–10% (*wt*/*wt*). P-THP is highly water soluble (>50 mg/mL), and its molecular size in an aqueous solution is 8.2 ± 1.7 nm [4].

### 4.2. Stability of P-THP and API Release

The release profile of the API (free pirarubicin) from the copolymer conjugate P-THP, its stability, protein and cell-binding profile, and solubility in various solutions have been thoroughly investigated [4,51]. Size exclusion chromatography of P-THP showed a hydrodynamic volume similar to that of bovine serum albumin (BSA) in an aqueous solution, with no apparent interactions with BSA or aggregation. The release of free pirarubicin was pH-dependent and confirmed at pHs 6.5 or lower, as shown in Figure 5 [4,51].

The release of the drug was significantly affected by the type of buffer used. Phosphate buffer seems to facilitate faster hydrazone bond cleavage at pH 7.4 whereas higher stability was achieved in an L-arginine solution which yielded only little cleavage and pirarubicin release (approximately 15% within 2 weeks) at the same pH at 25 °C [51]. As L-arginine has the potential to enhance the EPR effect [25], a solution containing L-arginine is an appropriate medium for formulating a P-THP infusion solution.

Furthermore, an ex vivo study using sera from different animal species showed very high P-THP stability. Incubation with blood also demonstrated high P-THP stability during circulation without binding to blood cells [51].

### 4.3. In Vitro Antitumor Activity

Cytotoxicity is five to ten times weaker with P-THP than with free pirarubicin probably because P-THP is internalized by tumor cells slowly, as with other macromolecules. However, P-THP itself demonstrates antitumor activity in various tumor cell lines, including the HeLa (cervical carcinoma), B16-F10 (mouse melanoma), HCT116 (human colon cancer), C26 (mouse colon cancer), U87-MG (human glioblastoma), and SUIT2 (pancreatic cancer) cell lines despite its stability and the absence of contamination by free pirarubicin in the aqueous solution [4,51,52,53]. The antitumor activity of P-THP is thought to result from the release of THP by the HPMA carrier because stronger cytotoxicity was observed when more THP was released in the presence of higher acidity [4]. However, P-THP itself also demonstrates antitumor activity, as seen in its relatively rapid cellular internalization in various experimental systems, including monolayer cells and cell spheroid [49,52,53]. P-THP also demonstrates cytotoxicity for various cell lines of neuroblastoma [54]. Although the antitumor activity was not enough for doxorubicin-resistant cell lines, the resistance was successfully reversed by adding a P-glycoprotein inhibitor. P-THP cytotoxicity for various pediatric tumor cell lines, including rhabdomyosarcoma, and Ewing sarcoma, is currently being researched.

### 4.4. Pharmacodynamics and In Vivo Antitumor Activity

A pharmacodynamic study using S-180 tumor-bearing mice investigated serial changes in body distribution of P-THP and free pirarubicin [4]. As Figure 6A,B show, P-THP persisted in the systemic circulation and significantly accumulated in tumor tissue at five hours after injection whereas free pirarubicin was cleared rapidly from the systemic circulation. The accumulation of P-THP was four to 20 times higher in tumor tissue than in normal tissue (except for the spleen) at 24, 48, and 72 h after administration (Figure 6B). The amount of pirarubicin released by P-THP into tumor tissue was highest at 24 h. Although the pirarubicin level gradually decreased (at 48 and 72 h), it was still much higher in the tumors than in any normal tissue (Figure 6C). Calculating the ratio of free pirarubicin to total pirarubicin (including P-THP) released demonstrated that the API continued to accumulate in the tumor tissue (Figure 6D), corroborating the observation that P-THP meets all three criteria described in the previous chapter for maximizing the effectiveness of nanomedicine.

As Figure 7 shows, the in vivo antitumor activity of P-THP was also investigated using the same murine model [4]. Pirarubicin or P-THP at 5 mg/kg and 15 mg/kg (the dosages were calculated as the pirarubicin equivalent) was administered via the tail vein in a single injection at ten days after S-180 cell inoculation when the tumor diameter was 5–8 mm. Although pirarubicin was effective at both levels and suppressed tumor growth throughout the experimental period (three months), two of the five mice which were given 15 mg/kg died from toxicity (Figure 7D). P-THP seemed more effective than pirarubicin at both dosages (Figure 7A,B), and all tumor-bearing mice treated with P-THP 15 mg/kg survived, with most showing tumor resolution by day 90 (Figure 7D). No weight loss or toxicity-related death was observed in this group (Figure 7C), suggesting that increasing the antitumor effect while concurrently reducing toxicity by altering the body distribution of pirarubicin, the API in this nanomedicine, can yield promising results.

### 4.5. Clinical Experience in Compassionate Use

As more preclinical data are published, patients with advanced-stage cancer may expect to receive P-THP even though it has not yet been approved. One such patient was a 60-year-old male with the diagnosis of prostate cancer with multiple metastases in the bilateral lungs, intrapelvic lymph nodes, soft tissue, femur, and sacrum. The treatment team first established an ethics committee comprising a bona fide third party, and the lead physician carefully explained the experimental nanomedicine to the patient and his family and obtained written consent. P-THP administration was begun after ethics committee approval was obtained.

The patient’s condition was refractory to treatment with leuprorelin and estradiol. After terminating the hormone therapy, he underwent proton-beam therapy with a 55 Gy equivalent dose targeting the primary prostate lesion. P-THP was first administered at a test dose of 30 mg/body (the THP-equivalent dosage) prepared in 200 mL of normal saline administered intravenously over 30 min without any acute adverse events. Then, the administration of a therapeutic dosage of 50 mg/body every 2–3 weeks was begun concurrently with the proton-beam radiotherapy. All the treatments were performed in the outpatient setting.

By the end of the treatment, his serum PSA level had normalized and continued decreasing below the lower evaluable limit. The multiple tumor metastases in the lungs had completely resolved on computed tomography seven months after the treatment, and the bone lesions had also resolved after 20 months of treatment. In terms of safety, the patient tolerated the entire course of therapy well and experienced no acute toxicities, such as mucositis, myelosuppression, abnormal blood chemistry values, cardiac toxicity, alopecia or gastrointestinal toxicity, including nausea and vomiting [55].

Although only one case is described here, our experience treating this patient demonstrated both the efficacy outcomes and excellent safety profile of P-THP. Our research team is currently planning for an early phase clinical trial of P-THP for refractory cancers, including breast cancer, ovarian cancer, prostate cancer, sarcomas of soft tissue (STS), and pediatric cancers.

## 5. Proposed Clinical Development of P-THP for Pediatric Solid Tumors

### 5.1. Premise for Applying P-THP to Pediatric Cancers

Most pediatric cancers are highly chemo-sensitive, and cytotoxic chemotherapy is always the mainstay of treatment. Among the cytotoxic agents, anthracyclines are the most frequently used in pediatric oncology and are highly effective against almost all pediatric cancers. Doxorubicin is basic to first-line treatment regimens for solid tumors, including neuroblastoma, Wilms tumor, hepatoblastoma, osteosarcoma, Ewing sarcoma, and non-rhabdomyosarcoma STS whereas daunorubicin is a fundamental ingredient of regimens for hematological malignancies, including acute lymphoblastic leukemia (ALL), acute myeloid leukemia (AML), and malignant lymphoma [2]. P-THP, therefore, may replace conventional anthracyclines in all first-line treatment regimens once its superior efficacy and safety profile are confirmed. Our discussion of the application of P-THP in this review is limited to solid tumors in which the EPR effect plays an important role. As pediatric solid tumors have high vascularization, a strong EPR effect enhancing P-THP efficacy may be expected [14].

The long-term survival rate in patients with pediatric cancers is reaching approximately 80%, but efforts should of course continue to be made to minimize the acute and chronic toxicities associated with anticancer treatments. Anthracyclines generate iron-mediated free radicals when metabolized by the mitochondria, as well as intercalate DNA by binding topoisomerase II, which is abundantly contained in cardiomyocytes. Both processes cause myocardial cell death and left ventricular systolic dysfunction [56]. Pirarubicin, the API in P-THP, is less cardiotoxic than doxorubicin; it has a cardiotoxicity of 0.62 relative to doxorubicin, and its maximum tolerated cumulative dose (MCTD) is 650 mg/m^2^ [57]. In Japan, pirarubicin is being studied for its efficacy against a variety of pediatric cancers and is expected to reduce the risk of cardiotoxicity of anticancer treatments. The safety and efficacy of pirarubicin have been evaluated in several clinical trials of treatments for neuroblastoma [41,42], hepatoblastoma [43], rhabdomyosarcoma [44], and pediatric leukemia [58]. As P-THP is expected to be much safer than pirarubicin due to the DDS mechanisms described above, it is very suitable for treating pediatric cancers and will contribute to reducing the acute and chronic side effects and long-term comorbidities, such as cardiac complications, associated with current treatments.

### 5.2. Strategic Choice of P-THP in Clinical Trials for Various Diseases

As multiagent cytotoxic chemotherapy is still the mainstay of multidisciplinary treatments for pediatric cancers, the potential role of P-THP depends on the disease. Multiple factors, including cancer subtypes, stage, disease status (e.g., newly diagnosed or refractory/recurrent), and comorbidities, must be accounted for when developing the optimal treatment strategy for each cancer subtype. Strategic considerations in P-THP development, summarized in Table 1, will be discussed in the following sections.

#### 5.2.1. Determining the Maximum Tolerated Dose (MTD) and Safety Profile

The first clinical trial should be a dose-finding study aimed at determining the MTD and safety profile of P-THP monotherapy in various types of pediatric cancer which are refractory to the standard treatments. There should be two cohorts; one with, and one without, a history of anthracycline-containing treatment. In cohort 1, the total amount of P-THP should be limited by the cumulative anthracycline dosage administered to each patient. The calculation of this limit should be based on the previously mentioned cardiotoxicity of pirarubicin relative to doxorubicin. In cohort 2, P-THP may be continued until the cumulative pirarubicin equivalent dosage reaches 650 mg/m^2^ although vigilance against cardiotoxicity is required. If the safety profile of P-THP is acceptable, the next step in its development may take two directions; the first of these is to conduct a dose-finding study to evaluate multiagent chemotherapy containing P-THP for use in subsequent trials. The other direction is to conduct a safety/efficacy study to evaluate maintenance treatments for refractory/metastatic cancers as will be discussed in the following section.

#### 5.2.2. Maintenance Treatment for Refractory/Metastatic Cancers

A trial setting in the early phase of drug development is very important for the treatment of pediatric cancers because of the current scarcity of treatment options for patients with very advanced disease. As P-THP is potentially less toxic to heavily treated patients, it may be a viable treatment option for such patients. Moreover, given that pirarubicin is a form of anthracycline, physicians may wish to test the efficacy of P-THP against anthracycline-sensitive cancers. However, testing P-THP in patients with recurrent or refractory cancer who have already been treated with anthracycline is ethically inappropriate, in view of the health risks posed by the cumulative dosage of anthracycline and the likelihood of a poor response, given the failure of previous treatments, including anthracyclines.

Therefore, for the purpose of developing a maintenance regimen, children with newly diagnosed metastatic STS, excluding rhabdomyosarcoma, may be chosen for enrollment. The rationale for such a choice is also bolstered by the fact that while doxorubicin is regarded as the standard treatment for metastatic STS, it is not a cure for this disease [59]. Once the safety profile of P-THP and its optimal dosage in monotherapy are established by dose-finding studies, all ethical criteria will have been met for this phase of drug development.

Usually, the cumulative dosage of anthracyclines must be capped at the individual MTCD. The MTCD of P-THP should be determined in the process of drug development. Patients with metastatic STS might wish to continue P-THP treatment if it is found to have an acceptable safety level and efficacy in controlling their disease on an individual basis. In this situation, researchers might determine the safety of cumulative dosages exceeding a 650 mg/m^2^ pirarubicin equivalent. If the regimen is found to be safe, another trial may be conducted to determine whether a P-THP maintenance regimen is superior to best supportive care in children with refractory/metastatic cancers.

#### 5.2.3. Replacement of Conventional Anthracyclines with a Less Toxic Treatment Regimen

Cardiovascular comorbidity secondary to anthracycline treatment may affect the survival outcome in pediatric cancers. A clinical trial of AML found that both event-free survival (hazard ratio: 1.6; *p* = 0.004) and overall survival (hazard ratio: 1.6, *p* = 0.005) were significantly worse in patients with cardiotoxicity [60]. Although dexrazoxane, a cardioprotective agent, significantly reduced the incidence of cardiovascular complications in cancer patients, it carries the risk of toxicities, including secondary malignancy [61]. From a safety point-of-view, replacing conventional anthracyclines with P-THP, which is at least equally effective theoretically while having fewer toxicities because of its selective distribution in the tumor site, may yield better efficacy outcomes due to the lower risk of adverse effects, including cardiotoxicity.

This very simple strategy, which is applicable to all pediatric tumors for which anthracyclines are used in the first-line treatment, may improve survival secondary to reducing the mortality rate associated with cardiac toxicity as suggested by the dexrazoxane experience [60]. Moreover, the high tumor selectivity of P-THP via the EPR effect has the potential to yield a higher antitumor effect resulting in better clinical outcomes. However, the potential benefits of this strategy should be tested for each cancer subtype in confirmatory randomized clinical trials, which usually require large cohorts and long-term follow up periods.

#### 5.2.4. Intensified Dose Intensity/Density Strategy

The dose intensity of doxorubicin may be an important determinant of favorable outcomes in patients with Ewing sarcoma or osteosarcoma [62]. A previous metanalysis found significantly better event-free and overall survival associated with anthracycline in patients with Wilms tumor and Ewing sarcoma [63]. Given the total dose of doxorubicin is limited by the risk of cardiac toxicity as described above, in recent years the efficacy and safety of dose intensification of alkylating agents, such as cyclophosphamide and ifosfamide, have been tested, especially in patients with Ewing sarcoma.

The clinical trial, INT-154 in the US [64], a randomized confirmatory study enrolling 478 patients with Ewing sarcoma, compared standard chemotherapy regimens with a vincristine, doxorubicin, and cyclophosphamide (VDC) regimen alternating with an ifosfamide and etoposide (IE) regimen for 48 weeks and intensified VDC-IE chemotherapy alternating with a high-dose cyclophosphamide and ifosfamide regimen for 30 weeks. Although this study failed to confirm the superiority of the intensified treatment partly because of differing treatment durations, investigating the potential for anthracycline dose intensification with P-THP to improve clinical outcomes is warranted.

The subsequent AEWS0031 trial, also in the US and enrolling 568 patients with localized Ewing sarcoma, used the strategy of interval compression, which involved administering VDC and IE every two weeks with intensive G-CSF support in the experimental arm [65]. This study demonstrated significantly higher five-year event-free survival in the experimental (interval compressed) arm (73% in experimental arm vs 65% in the standard arm, *p* = 0.048) without increased toxicity. Repeating chemotherapy every two weeks increases hematological toxicity; incorporating novel, less toxic drugs can therefore improve the feasibility of this strategy. P-THP, which is possibly less myelotoxic than doxorubicin, should be considered a less toxic candidate drug for Ewing sarcoma.

#### 5.2.5. Add-On to Standard Chemotherapy without Anthracyclines

The role of anthracyclines in neuroblastoma, which still has a poor prognosis, is controversial. Although anthracyclines are considered a part of standard regimens for high-risk neuroblastoma in the US and Japan, doxorubicin use is avoided in first-line chemotherapy regimens containing cisplatin, vincristine, carboplatin, etoposide, and cyclophosphamide (rapid COJEC) in the EU [66]. The reason for this avoidance derives from a study by Shafford, et al., which found no improvement in the treatment response rate in neuroblastoma after the addition of doxorubicin every three weeks to induction chemotherapy with vincristine, cisplatin, epipodophyllotoxin (VM26), and cyclophosphamide (OPEC) [67].

The role of anthracyclines in rhabdomyosarcoma treatment has also been controversial since the 1970′s. Despite the significant activity of doxorubicin against both newly diagnosed and recurrent RMS, Intergroup Rhabdomyosarcoma Study (IRS) I and II both demonstrated no additive effect of doxorubicin in the standard vincristine, actinomycin-D, and cyclophosphamide (VAC) chemotherapy regimen [68,69]. Recently, RMS2005 addressed the very same issue in the ifosafamide-based chemotherapy but failed to prove the superiority of additional doxorubicin [70]. ARST0431, which incorporated biweekly VDC-IE as the chemotherapy backbone in a complicated, six-drug, combination regimen [71], demonstrated only minimal efficacy.

In these studies, the additional toxicities of doxorubicin, such as cytopenia and mucositis, might have delayed the subsequent treatment course and increased adverse events affecting the clinical outcomes. The effectiveness of adding anthracycline (i.e., P-THP) to the current standard regimen in improving clinical outcomes in neuroblastoma and rhabdomyosarcoma should also be tested.

## 6. Concluding Remarks

The aim of multidisciplinary treatments for pediatric cancers is always to effect a cure even when the disease is at an extremely advanced stage. As repeatedly stated in this review article, most pediatric cancers are so chemo-sensitive that the role of cytotoxic agents is much more important than in the treatment of adult cancers. Anthracyclines are highly effective for almost all pediatric cancers, and P-THP may be expected to augment their efficacy once the appropriate dosage is confirmed in early-phase clinical trials. Although the process of clinical development described in the previous chapter is largely speculative, we may expect P-THP to confer more benefits for the treatment of pediatric cancers than precision medicine, which is currently dominating oncological research.

The history of pediatric oncology tells us that only individually-tailored multidisciplinary treatment has a high probability of effecting a cure in pediatric cancers. A single, therapeutic agent, however excellent it may be, cannot replace multidisciplinary treatment aimed at eradicating the cancer. Moreover, intra-tumoral heterogeneity and associated drug resistance in the course of a disease can also modulate the treatment response, especially to molecular targeting agents [72]. Therefore, continued, parallel development of cytotoxic agents and molecular targeting agents is warranted. We hope that P-THP will usher in a new era of “nontoxic” cytotoxic agents and prove to be a highly effective weapon against pediatric cancers.

## Figures and Tables

**Figure 1 cancers-13-03698-f001:**
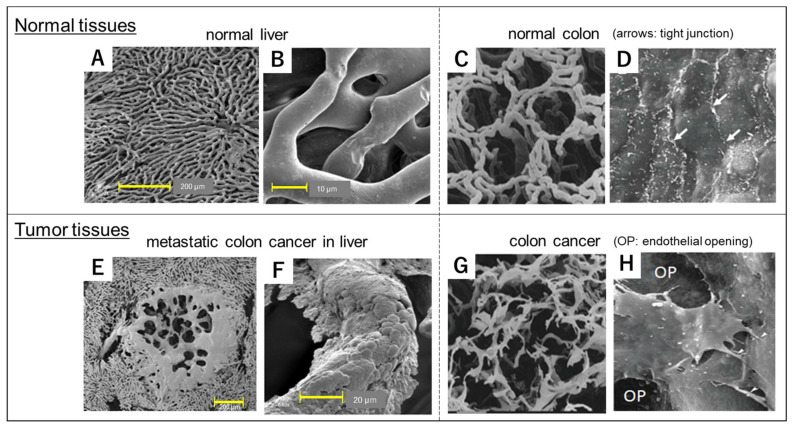
Comparison of scanning electron microscopy (SEM) images of blood vessels in normal, healthy tissues (**A**–**D**) and tumor tissues (**E**–**H**). Blood vessels in healthy tissue in (**A**–**C**) show clear, smooth, regular features and no leakage of polymeric resin. In contrast, the tumor vessels show leakage of polymeric resin at the capillary level (**E**). Although normal colonic tissue consists of organized vascular casts (**C**), colon tumor tissue shows a disorganized, irregular vascular network (**G**). The luminal surface of normal blood vessels (**D**) shows tight cell-cell junctions in the endothelium whereas blood vessels in the tumor (**H**) have large gaps (OP in **H**) among the endothelial cells. Adapted from reference [14] with permission. Images (**C**,**D**,**E**,**H**) are courtesy of Professor Moritz Anton Konerding in Mainz, Germany.

**Figure 2 cancers-13-03698-f002:**
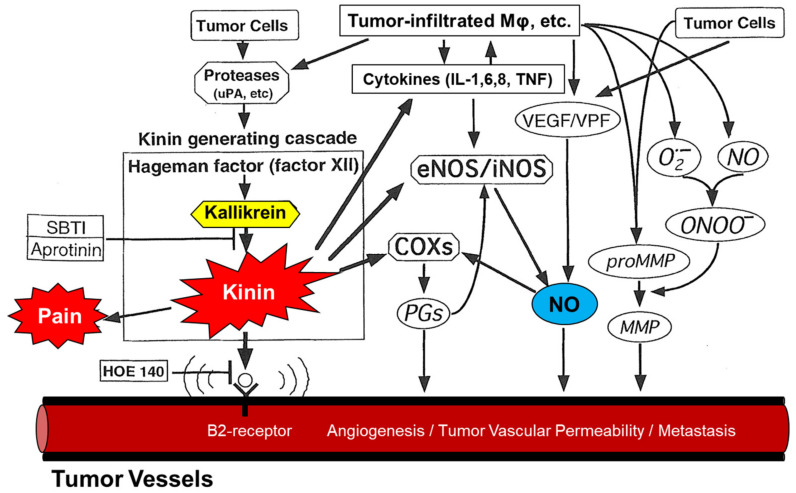
The enhanced permeability and retention (EPR) effect in tumor vasculature. The mechanism of this tumor-selective macromolecular drug targeting depends on various effectors affecting vascular tone as shown here. Aprotinin is an inhibitor of kallikrein; HOE-140 is a peptide antagonist of kinin. SBTI, soybean trypsin inhibitor; NO, nitric oxide; eNOS, endothelial nitric oxide synthase; iNOS, inducible form of nitric oxide synthase; COXs, cyclooxygenases; PGs, prostaglandins; MMP, metalloproteinase; ONOO^-^, peroxynitrite; O_2_^−^, superoxide anion radical; MF, macrophage; VEGF, vascular endothelial growth factor; VPF, vascular permeability factor; uPA, urokinase plasminogen activator; IL, interleukin; TNF, tumor necrosis factor; B2 receptor, bradykinin B2 receptor. Adapted from ref. [3].

**Figure 3 cancers-13-03698-f003:**
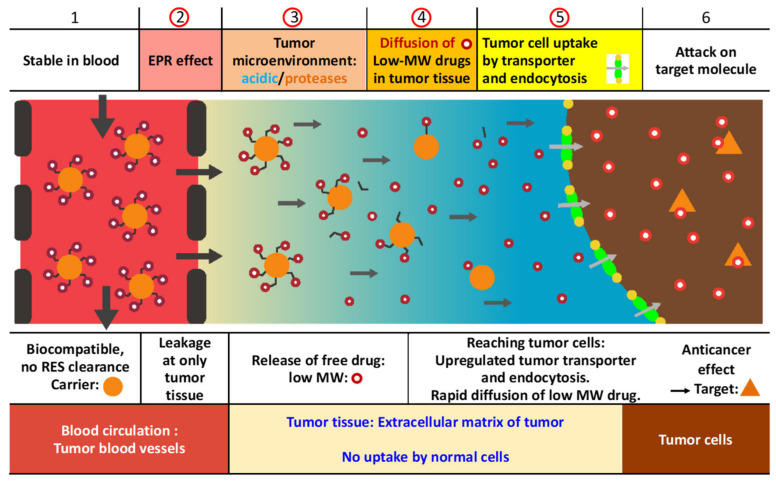
Multiple barriers and necessary conditions for overcoming them in nanomedicine targeting tumors. To maximize the effectiveness of nanomedicine, the following three conditions must be met: (**1**) Selective accumulation of the macromolecule in the tumor; (**2**) release of the active pharmaceutical ingredients (API) into the tumor tissue; and (**3**) active cellular uptake of the API into the tumor cells. Adapted from reference [29].

**Figure 4 cancers-13-03698-f004:**
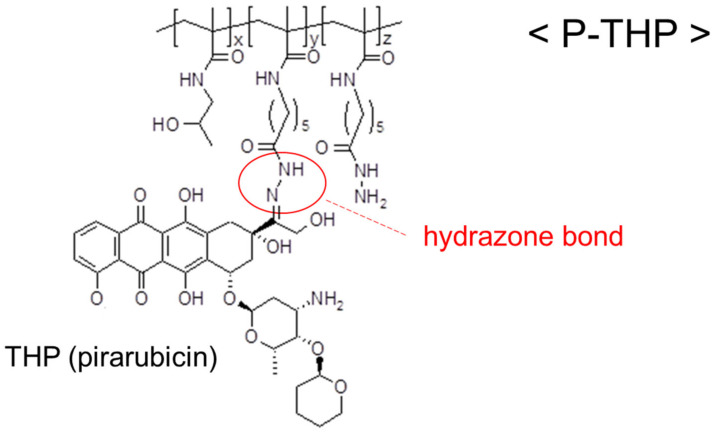
Chemical structure of hydroxypropyl-acrylamide polymer-conjugated pirarubicin (P-THP). Pirarubicin, the active pharmaceutical ingredient (API), is connected to hydroxypropyl-acrylamide (HPMA) polymer by the hydrazone bond, an acid-cleavable linkage. The structure demonstrates the enhanced permeability and retention (EPR) effect targeting the tumor and the secondary release of the API there exerting the antitumor effect.

**Figure 5 cancers-13-03698-f005:**
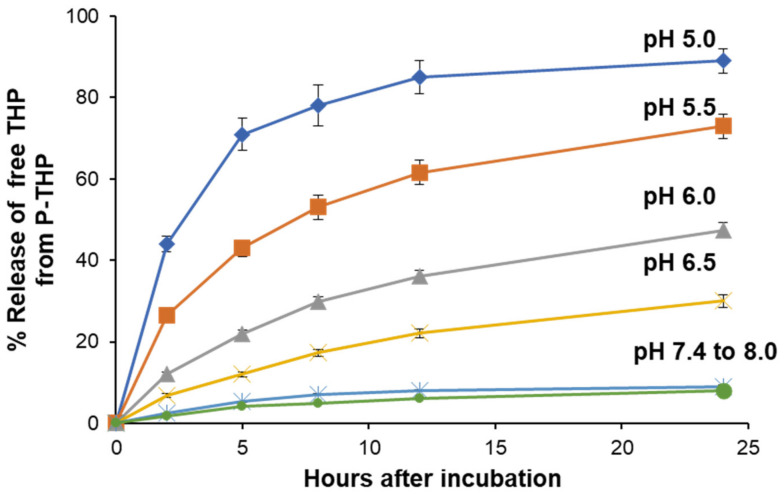
Release profile of free pirarubicin (THP) from the polymer conjugate (P-THP) at 37 °C. P-THP was dissolved in buffer solutions with different pH. The release of THP was determined by HPLC. Adapted from reference [51].

**Figure 6 cancers-13-03698-f006:**
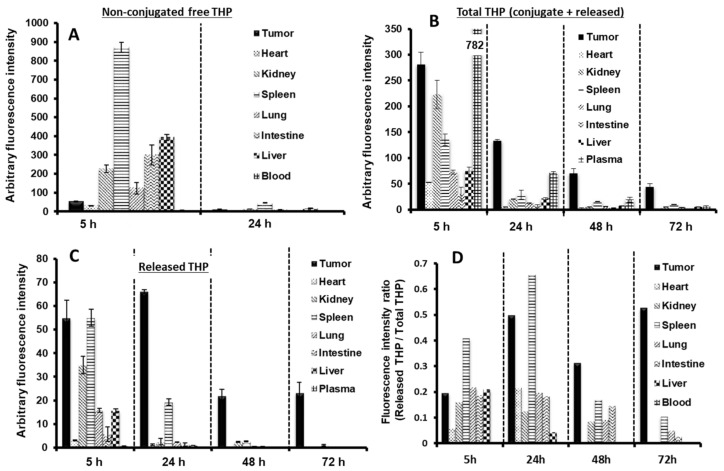
Body distribution of P-THP. (**A**) Free pirarubicin (THP) was administered at 10 mg of THP per kg equivalent. (**B**) and (**C**) Profile of P-THP administered at 10 mg of THP per kg equivalent into S-180 tumor-bearing mice. At the indicated time periods, mice were anesthetized and tissues were collected. (**B**) Total THP content and (**C**) released free THP content in each tissue sample were measured using HPLC. Values are expressed as the mean ± S.E. (*n* = 3). (**D**) Ratio of free THP to total THP in each tissue. The fluorescence intensity of the free THP was divided by the fluorescence intensity of the total THP. Adapted from ref. [4].

**Figure 7 cancers-13-03698-f007:**
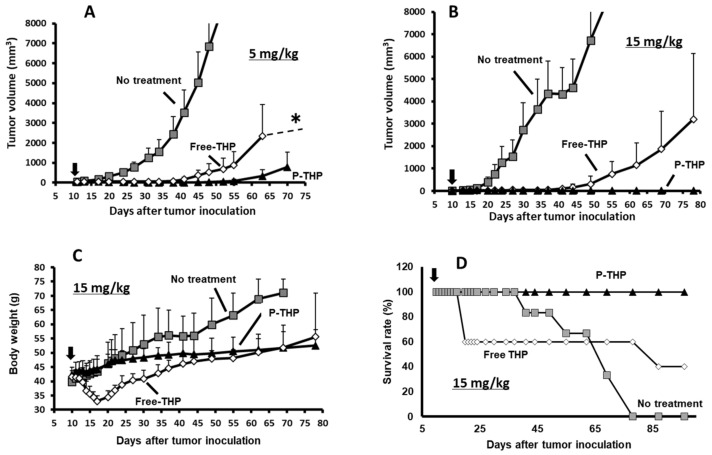
Antitumor activity of P-THP in vivo. (**A**,**B**) P-THP or free pirarubicin (THP) was injected once at 5 mg/kg (**A**) or 15 mg/kg (**B**) of THP per kg equivalent into S-180 tumor-bearing mice. (**C**) Body weight change and (**D**) survival rate after administration of 15 mg/kg of THP per kg equivalent to S-180 tumor-bearing mice. Values are expressed as the mean ± S.E. (*n* = 5–6). * One of the five mice with the largest tumor died; the tumors in the remaining mice continued to grow. Adapted from reference [4].

**Table 1 cancers-13-03698-t001:** Clinical development of P-THP for pediatric solid tumors.

Phase	Disease/Status	Primary Aim	Design
1	Miscellaneous/recurrent	To determine MTD and safety profile of P-THP monotherapy	Rolling-six dose-escalation
	Cohort 1 (h/o anthracyclines +)		
	Cohort 2 (h/o anthracyclines −)		
1–2	Miscellaneous/recurrent	To determine MTD and safety profile of combination therapy containing P-THP	Rolling-six dose-escalation plus extension cohort
2	Non-rhabdomyosarcoma STS	Safety/efficacy evaluation of P-THP monotherapy	1-arm, exploratory
	Miscellaneous/newly diagnosed	Safety/efficacy evaluation of combination therapy	wP2 design, etc.
3	Miscellaneous/recurrent	Efficacy confirmation of P-THP monotherapy	RCT w/ BSC
	Hepatoblastoma, Wilms Tumor, etc.	Superiority confirmation of P-THP replacement	RCT w/dox regimen
	Osteosarcoma, Ewing sarcoma	Superiority confirmation of P-THP regimen (possibly w/intensification of anthracycline)	RCT w/dox regimen
	Neuroblastoma, Rhabdomyosarcoma	Superiority confirmation of P-THP add-on	RCT w/std regimen

BSC; best supportive care, h/o; history of, MTD; maximal tolerating dose, P-THP; hydroxypropyl acrylamide polymer-conjugated pirarubicin, RCT; randomized controlled trial, STS; soft tissue sarcoma, w/; with, wP2; window phase 2.

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
