# Peer review of "Development of a Selective Tumor-Targeted Drug Delivery System: Hydroxypropyl-Acrylamide Polymer-Conjugated Pirarubicin (P-THP) for Pediatric Solid Tumors"

_cancers, 2021, doi:10.3390/cancers13153698_

Round 1

Reviewer 1 Report

The paper of Makimoto and colleagues is very well written and structured and the P-THP holds for promise to be introduced in cancer therapeutic protocols. On the other hand, pediatric cancer raises a medical and ethical conundrum that has to be solved as fast as possible. In this regard and having in mind the issue of the journal, I believe that the authors missed the goal and that unfortunately, the present work does not provides enough data of interest to be published at least inside the topic of pediatric cancers. Some of my main concerns are listed below:

  • The tittle is not appropiate for a review and is confusing.
  • Points 2,3 about the EPR effect are interesting and very well described and historically, the contribution of H Maeda on this topic is huge. I can understand that many of the figures are reprints but H. Maeda et al.  just published a paper on this year on the same topic, so why also here again if there is no connection with pediatric cancers in terms of studies performed in the pediatric population at least? 
  • The authors have to concrete along the text (includying the title and line 471 for example) that this approach is aimed for some pediatric solid tumors and that it could not be interesting for blood tumors and not even CNS tumors, which by the way represent the vast majority of pediatric cancers.
  • There is no in vitro and in vivo data using pediatric cancer tumor models.
  • Point 4.5. Children are not just small adults and most of the pediatric solid cancers have a big genetic burden so any comparison with a compassionate use on an 60 years-old adult seems speculative.
  • In general there is an evident lack of studies/references concerning the topic of pediatric cancer and many statements are based on personal assumptions and no proofs. This should not be suitable for an article review. 
  • Chemotherapeutic toxicity in pediatric patients should be explored in depth. The point 5.1 is a good start.

Author Response

First, the authors, including Dr. Hiroshi Maeda, thank the reviewer very much for the valuable suggestions on improving the manuscript.

The purpose of this review article is to discuss the clinical development of a specific nanomedicine (P-THP) for pediatric solid tumors based on our experience with adult cancers and the methodology of clinical development rather than to describe the role of nanomedicine in pediatric oncology, which is outside our scope. In this regard, the present review article is not a repetition of another article by Maeda (reference #3), which comprehensively describes the history of nanomedicine over the past 35 years and its prospects for the future. Moreover, we should emphasize that there is a connection between P-THP and pediatric cancers (at least pediatric solid tumors including CNS tumors) based on the three conditions described below.

  • As we consistently mention in the article, most pediatric cancers are highly chemo-sensitive (in general, much more sensitive than adult cancers, including prostate cancer) regardless of the presence of a genetic burden (which is provable in less than 10% of pediatric tumors). Anthracyclines are among the “key” drugs for the treatment of most forms of pediatric solid tumor, as mentioned in detail throughout chapter 5. Because the cytotoxic effect of P-THP is exerted through its API (pirarubicin), we can expect P-THP to be effective against pirarubicin-sensitive pediatric tumors, including neuroblastoma, hepatoblastoma, and rhabdomyosarcoma, as we mention in lines 396-398.
  • As described in lines 385-387, pediatric solid tumors generally have high vascularization. Therefore, a strong EPR effect enhancing P-THP efficacy may be expected. The rationale for this speculation is based on basic experiments using Evans-blue albumin (EBA) in murine tumor models in which EBA accumulation was observed only in hypervascular regions of the tumors (reference #14). We added the reference #14 after the sentence. In terms of hypervascularity, pediatric CNS tumors are not the exception, as tumors show a disorganized blood-brain barrier. We believe that anthracycline-sensitive CNS tumors, such as atypical teratoid / rhabdoid tumors, are good targets.
  • We do have in vitro data for pediatric cancers but were not able to include them for lack of an appropriate context in the manuscript. Following your advice, we added the data in lines 297-300 as below.

P-THP also demonstrates cytotoxicity for various cell lines of neuroblastoma [56]. Although the antitumor activity was not enough for doxorubicin-resistant cell lines, the resistance was successfully reversed by adding a P-glycoprotein inhibitor.  [56] Koziolová et al. J Control Release 233, 136-146, 2016.

We believe that our article now contains enough information to discuss the clinical development of P-THP for pediatric solid tumors in accordance with methodology of clinical development of novel agents. However, the title is possibly misleading as the reviewer pointed out. We changed the words “pediatric cancers” to “pediatric solid tumors”. We also removed “highly” because Reviewer 3 thought that it might be an exaggeration.

Development of a Selective Tumor-Targeted Drug Delivery System: Hydroxypropyl-Acrylamide Polymer-Conjugated Pirarubicin (P-THP) for Pediatric Solid Tumors

In terms of the description in line 471 (lines 468-471 in the current version), which is related to AML (hematological malignancy), the authors believe that the fact (reducing cardiotoxicity resulted in improved overall survival) can be extrapolated to pediatric solid tumors and that the hypothesis should be tested in clinical trials.

As the reviewer comments, we have also given much thought to the long-term safety of novel agents for the pediatric population. We have carefully described our perspective in chapter 5.

Reviewer 2 Report

The review by Makimoto et al. comprehensively describes the background and significance of development of the P-THP for pediatric solid cancers, including a proposal on potential clinical trial. The manuscript includes the substantial review of the field, including the EPR effect described by one of the co-authors (an expert in the field Dr. Hiroshi Maeda). It is very well described and discussed why the use of this drug has a great potential for pediatric solid cancers. There are though a few critiques that can be useful to consider:

1) Although it is very important to acknowledge the role of EPR in nanomedicine, it feels as the description of its discovery, basis, and mechanism can be condensed into a smaller sub-chapter. EPR was discovered in 1986 and has become a fundamental knowledge to researchers in the field. Chapters 1-3 take away the focus away from the development of nanoformulated pirarubicin for childhood cancers.

2) There are redundancies that can be avoided, such as Chapter 4 (1st paragraph, page 6) lists 3 criteria for efficacy of nanomedicine that have been already listed in Chapter 3 (1st paragraph, page 4). Referencing the list of criteria in one of these chapters should be sufficient. Additionally, the criteria in both Chapters reference different publications (#30 and #4). 

3) Chapter 5 probably needs to have a word "proposed' before the title as the strategic development of a clinical trial is only proposed at this stage.

4) There is a reference to Table 1 in text (page 12), while the only table in the manuscript is "Table 2". 

5) Chapter 5.2.3 describes the justification of usage of P-THP as a less toxic drug - this has been mentioned and presented in previous chapters. This sub-Chapter seems redundant.

Overall, this is a good review of the current basic knowledge and state of nanomedicine, with the focus on P-THP for pediatric cancers as a future avenue.

Author Response

The authors appreciate the important comments. Below are our replies to each of your comments.

  • We understand that there is a risk that Chapters 2 and 3 (excluding Chapter 1 because it is the “Introduction”) take away focus from the development of P-THP for childhood cancers. However, if we try to combine these two chapters, the result would be too long and confusing for readers (especially if they are clinicians) to understand the mechanism of the EPR effect and its clinical application. That is why we kept these chapters separate. We believe that the current structure, with the sequence from general (Chapter 3; nanomedicine) to specific (Chapter 4; P-THP), is therefore appropriate.
  • As pointed out, there is a redundancy. We omitted the description of the criteria from Chapter 4 (paragraph 1, page 6). We also combined the references into #30 in both parts.
  • We added “proposed” to the title of Chapter 5.
  • We corrected the title of the table to “Table 1”.
  • Chapter 5.2.3 elaborates not only the safety of P-THP but also the close relationship of the true endpoint of the clinical trial (overall survival) to increased safety of the treatment. There is no similar description in the other chapters, so we do not think it is redundant.

Reviewer 3 Report

Cancers-1263980 "Development of a highly-selective tumor-targeted drug delivery system: Hydroxypropyl-acrylamide polymer-conjugated pirarubicin for pediatric cancers" is described that P-THP is effective and reduces adverse effects for pediatric solid tumors. 

To highlight the application of P-THP for pediatric cancers, please mention several points as following questions.   

  1. Developed P-THP is a passive targeting nanocarrer using EPR effect. Active targeting drug delivery system is higher selective than passive targeting. So, I thought the expression "highly selective tumor target" is exaggerated in the title of this manuscript. I wonder that Active targeting THP is not developed yet? What is the limitation of P-THP? If the limitation of P-THP is, describe it in Discussion, please.  
  2. Cancers are various properties depending on original tissues. Nevertheless, authors described the application of P-THP to pediatric cancers (Almost all types) is available. What is the merits of THP compared to other anticancer drugs for the treatment of pediatric cancers? Please, discuss in more detail the reason for selecting THP for pediatric cancers. 

[minor]

  1. at line 21, API is correct? not Apis?

Author Response

The authors thank Reviewer 3 for the useful advice on improving our manuscript.

Following the advice, we removed “highly” from the title. Also, we changed “pediatric cancers” to “pediatric solid tumors” in accordance with the comment by Reviewer 1.

One limitation of P-THP is possibly the potential for cardiotoxicity when the cumulative dose exceeds 650 mg/m2 pirarubicin equivalent. This problem is discussed in Chapter 5.2.1 and 5.2.2 on page 12. Another concern is tumor resistance to anthracyclines, which we mention in the additional description in lines 297-300 on page 8:

P-THP also demonstrates cytotoxicity for various cell lines of neuroblastoma [56]. Although the antitumor activity was not enough for doxorubicin-resistant cell lines, the resistance was successfully reversed with addition of a P-glycoprotein inhibitor.  [56] Koziolová et al. J Control Release 233, 136-146, 2016.

Honestly, we have very little clinical experience using P-THP in humans. Therefore, describing clinical safety data in detail is almost impossible, and discussing our speculations may be misreading.

On the other hand, the merits of THP (pirarubicin) have been described in Chapter 5.1, page 11 as: (1) Anthracyclines are the most frequently used drugs in pediatric oncology and are highly effective against almost all pediatric cancers. (2) Pirarubicin is less cardiotoxic than doxorubicin; it has a cardiotoxicity of 0.62 relative to doxorubicin, and its maximum tolerated cumulative dose is 650 mg/m2. (3) The safety and efficacy of pirarubicin have actively been evaluated in several clinical trials of treatments for neuroblastoma, hepatoblastoma, rhabdomyosarcoma, and pediatric leukemia.

We corrected the typos (or misconversion by WORD) from Apis to APIs in the sections 3.3 and 4.1.

Reviewer 4 Report

An interesting discussion on the EPR effect and how this may guide future therapies for childhood cancers. The manuscript is well written, there is author citation, but in this case I believe it is valid. The work draws on the years of expertise of the group and uses this to authoritatively make judgements on potential of future therapies. I believe this is one of those rare occasions when there is nothing which could improve the article, and I suggest to accept in current form.

Author Response

We appreciate your review of our article. We modified several parts of the manuscript according to the advice we have received from the other reviewers. We hope that the changes have improved the quality of our article.

Round 2

Reviewer 1 Report

-

Reviewer 3 Report

The revised manuscript is written according to the comments of reviewer.